# Parameterization and Remote Sensing Retrieval of Land Surface Processes in the Gurbantunggut Desert, China

Wei Li [1,2,3,4,5,6], Jiacheng Gao [2,3,4,5,6], Ali Mamtimin [2,3,4,5,6,*], Yongqiang Liu [1], Yu Wang [2,3,4,5,6], Meiqi Song [2,3,4,5,6], Cong Wen [2,3,4,5,6], Ailiyaer Aihaiti [2,3,4,5,6], Fan Yang [2,3,4,5,6], Wen Huo [2,3,4,5,6], Chenglong Zhou [2,3,4,5,6], Wenbiao Wang [7] and Zhengnan Cui [7]

1    College of Geography and Remote Sensing Sciences, Xinjiang University, Urumqi 830046, China
2    Institute of Desert Meteorology, China Meteorological Administration, Urumqi 830002, China
3    National Observation and Research Station of Desert Meteorology, Taklimakan Desert of Xinjiang, Urumqi 830002, China
4    Taklimakan Desert Meteorology Field Experiment Station of CMA, Urumqi 830002, China
5    Xinjiang Key Laboratory of Desert Meteorology and Sandstorm, Urumqi 830002, China
6    Key Laboratory of Tree-Ring Physical and Chemical Research, China Meteorological Administration, Urumqi 830002, China
7    Elion Resources Group Co., Ltd., Beijing 100026, China
*    Correspondence: ali@idm.cn

**Abstract:** The exchange of energy between the land surface and atmosphere is dependent upon crucial parameters, including surface roughness, emissivity, bulk transfer coefficients for momentum ($C_D$) and heat ($C_H$). These parameters are calculated through site observation data and remote sensing data. The following conclusions are drawn: (1) the aerodynamic roughness of the Gurbantunggut Desert measures $1.1 \times 10^{-2}$ m, which is influenced by the varying conditions of the underlying surface. The roughness decreases as wind speed increases and is seen to be directly proportional to the growth of vegetation. From April to June, the aerodynamic roughness increases with increasing vegetation cover, but begins to gradually decrease after July. Spatially, the middle regions show higher roughness values than the eastern and western areas. In the central part of the desert, the roughness is between $2.37 \times 10^{-2}$ m and $2.46 \times 10^{-2}$ m from April to November. The northwest and northeast regions measure $1.41 \times 10^{-2}$ m–$2.04 \times 10^{-2}$ m and $1.53 \times 10^{-2}$ m–$2.39 \times 10^{-2}$ m, respectively. (2) The surface emissivity is 0.93, and it varies depending on the snow and vegetation present in the underlying area. (3) $C_D$ and $C_H$ exhibit an inverse relationship with wind speed. When wind speed falls below 6 m/s, the $C_D$ declines rapidly as wind speed increases. In contrast, once wind speed surpasses 6 m/s, the propensity for the $C_D$ to decrease with increasing wind speed slows down and approaches stability.

**Keywords:** Gurbantunggut Desert; remote sensing retrieval; aerodynamic roughness; surface emissivity; bulk transfer coefficients for momentum ($C_D$) and heat ($C_H$)

## 1. Introduction

The Gurbantunggut Desert, the largest semi-fixed desert in China [1,2], boasts a distinctive climate with remarkable seasonal variations. While winter brings long-term snow, spring marks the emergence of quick-witted and hardy plants. This unique desert environment plays a key role in shaping regional climate formation. The climate system is mainly driven by the exchange of materials and energy between land and air. The key parameters for energy exchange include surface roughness [3], surface emissivity, and bulk transfer coefficients for momentum ($C_D$) and heat ($C_H$). Therefore, scrutinizing these parameters and their influencing factors is crucial for understanding land–air interaction in desert areas.

A large number of studies have been carried out on the surface parameters, atmospheric boundary layer, and stability of each underlying surface in the arid and semi-arid zone [4], and Peng et al. [5] combined machine learning techniques, wind profile equations, station observations, and MODIS remote sensing data to estimate the daily dynamic roughness on a global scale. Trepekli et al. [6] used a UAV-mounted lidar system to estimate the dynamic roughness of the underlying surface under farmland. Ma et al. [7] evaluated a variety of thermodynamic roughness schemes, and the results showed that the C97 scheme had the best effect on the underlying surface in the grassland of eastern Tibet. Yang et al. [8] found that the thermodynamic roughness changed significantly between day and night, and the average value did not change significantly with the kinetic roughness, and compared the $kB^{-1}$ and specific emissivity calculated by the seven schemes. Yang Aqiang et al. [9] used MODIS product data to estimate the aerodynamic roughness and zero-plane displacement height in eastern China and found that they had seasonal variation characteristics. Wang et al. [10] used the observation data of the SACOL (Semi-Arid Climate and Environment Observatory of Lanzhou University) station to calculate the dynamic roughness and overall transmission coefficient of the station and found that the change of stability significantly affects the momentum and $C_H$ and the change of the transmission coefficient caused by stability in one day can exceed the seasonal change caused by the change of the underlying surface. Wang et al. [11], based on the Terra-MODIS L3 grade product MOD11C3, analyzed the temporal and spatial changes of the specific emissivity of China and concluded that the spatiotemporal distribution of the specific emissivity in China is related to temperature, and the higher the specific emissivity, the lower the temperature. Wang et al. [12] used the 11 flux observations of the third test of the Qinghai-Tibet Plateau (TIPEX III) to analyze the land surface parameters and turbulence characteristics of the Qinghai-Tibet Plateau and its surrounding areas and concluded that, under unstable conditions, the relationship between the momentum $C_H$ and the 10 m wind speed obeys the power law, and under stable conditions, when the wind speed is less than 5 m/s, the momentum $C_H$ increases with the increase in wind speed. Several researchers used the wind speed profile method and wind tunnel simulation to study the dynamic roughness of a flat sandy bed and flat sand [13–16]. Liu et al. [17] calculated multiple surface parameters in the hinterland of the Taklamakan Desert and found that the dynamic roughness in the region did not vary seasonally, mainly depending on wind speed. The $C_H$ is higher in winter and lower in summer. Most of the previous studies on land surface parameters focused on grassland, forest, and arable land and less on desert areas, which are extremely sensitive to climate change response because of their special environment and directly affect human activities in oasis areas. The research focuses on the Gurbantunggut Desert and uses 2017 data from the eddy covariance system. It examines the key parameters of energy exchange and factors that influence land–air interaction. Additionally, MODIS data are used to retrieve aerodynamic roughness as a supplement to existing research. This fills in some gaps and provides a necessary reference for studying land–air material energy exchange and environmental protection in desert areas.

## 2. Materials and Methods

### 2.1. Data and Information

#### 2.1.1. Site Data

The Gurbantunggut Desert is located in the middle of the Junggar Basin in Xinjiang, China, with a total area of about $4.88 \times 10^4$ km², accounting for about 6.8% of the desert area of the country. It is the second largest desert in China. The average annual precipitation is less than 150 mm, rainfall is mainly concentrated in spring, and the surface is usually covered with stable snow about 15 cm depth in winter. The spring snow melt and precipitation is more, and the soil moisture content of sand dunes reaches the highest throughout the year, which creates good conditions for the growth and development of short-lived vegetation. In this paper, the surface parameters and characteristics of the Gurbantunggut Desert analyzed by 2017 observation data from the Land-Atmospheric

Interaction Observation Station in the Ke La Mei Li area (hereinafter referred to as KLML Station) of the hinterland of the Gurbantunggut Desert, which was established by the Institute of Desert Meteorological, China Meteorological Administration, Urumqi in 2012.

The KLML station (45.914′08.19″N, 87.935′23.79″E, 531 m above sea level) is located in the middle of the Gurbantunggut Desert, and its administrative boundaries belong to the Ke La Mei Li area of Kalamagay Township, Fuhai County, Altay Region, Xinjiang, China. The terrain around the site is flat and open, and the height of the dunes is generally below 50 m. The data used in this paper are from the eddy covariance system, which uses an open-path infrared gas analyzer (Model Li7500A, Licor, Lincoln, NE, USA) and a 3Dsonic anemometer (Model Gill Wind Master Pro, GILL Instruments Ltd., LMT, Coventry, UK) with an installation height of 10 m. Before calculating the parameters, the eddy data was strictly controlled using the EddyPro 7.0.6 software launched by LI-COR in the United States.

### 2.1.2. Remote Sensing Data

The satellite data comes from MODIS, which is an important sensor in the satellites TERRA and AQUA launched by the US Earth Observation System (EARTH) program. The band of the MODIS sensor covers the full spectrum from visible light to thermal infrared, which can detect surface and atmospheric conditions such as surface temperature, surface vegetation cover, and atmospheric precipitation, with a maximum spatial resolution of 250 m. The normalized vegetation index obtained by MODIS detection is from MODIS's MOD13Q1 product, which provides a global image with a spatial resolution of 250 m every 16 days; 2017 data from the Gurbantunggut Desert was selected.

### *2.2. Site Parameter Calculation*
### 2.2.1. Aerodynamic Roughness

Aerodynamic roughness is the roughness of the ground [18], which is the height at which the wind speed is zero on the logarithmic wind profile near the ground. In this paper, using the measured data of the KLML station, according to the Monin–Obukhov similarity theory, the aerodynamic roughness is calculated as follows:

$$\ln z_{om} = \ln z_m - \varphi_m - kU/u_* \tag{1}$$

$$u_* = \left[ \left( \overline{u'w'} \right)^2 + \left( \overline{v'w'} \right)^2 \right]^{\frac{1}{4}} \tag{2}$$

where the Kármán constant $k = 0.4$, the gravitational constant $g = 9.8$ m·s$^{-2}$, and $z_m$ is the installation height of the eddy covariance system—in this paper, $z_m = 10$; $U$ and $u_*$ are the horizontal wind speed and frictional wind speed at $z_m$ height—the former is observed by the anemometer, and the latter is calculated by the three-dimensional wind speed observed by the anemometer, both in m/s. The atmospheric stability conditions are judged by $(z–d)/L$, $d$ is the zero-plane displacement height, and $\varphi_m$ is the dimensionless function of atmospheric stability, which is zero under neutral atmospheric stability conditions and is calculated by Equations (3) and (4) under unstable atmospheric conditions and stable conditions, respectively:

$$\varphi_m \left( \frac{z_m}{L} \right) = \ln \left( \frac{1+x^2}{2} \right) + 2\ln \left( \frac{1-x}{2} \right) + \tan^{-1}(x) + \frac{\pi}{2} \tag{3}$$

$$\varphi_m \left( \frac{z_m}{L} \right) = -5.3 \left( \frac{z_m}{L} \right) \tag{4}$$

$$L = \frac{T_a u_*^{\,2}}{kg T_*} \tag{5}$$

$$T_* = -\mathrm{H}/\left(\rho c_p u_*\right) \tag{6}$$

where $x = [1-19z_m/L]^{1/4}$; $L$(m) and $T_*$(K), respectively, are the Monin–Obukhov rough length and friction temperature, which can be calculated from Equations (5) and (6). The screening wind speed is greater than 1 m/s; the friction speed is greater than 0.01 m/s; the sensible heat flux is greater than 10 W/m$^2$; no rain; the acquisition time is daytime flux data.

Thermodynamic roughness is usually described using heat-transfer additional damping kB$^{-1}$ as follows [8]:

$$\mathrm{kB}^{-1} = \ln(z_{om}/z_{oh}) \tag{7}$$

$$z_{oh} = (70v/u_*)\cdot \exp\left(-\beta u_*{}^{0.5}|T_*|^{0.25}\right) \tag{8}$$

In the formula, the $v$-air flow adhesion coefficient is $1.5 \times 10^{-5}\mathrm{m^2 \cdot s^{-1}}$; $\beta = 7.2\ \mathrm{m^{-0.5} \cdot s^{0.5} \cdot K^{-0.25}}$.

### 2.2.2. Surface Emissivity

Surface emissivity refers to the ratio of the radiation capacity of an object to the radiation capacity of a black body at the same temperature. In this paper, the surface emissivity is calculated according to the physics-based semi-empirical method of Yang et al. [8], and the formula is as follows:

$$R_{lw}^{\uparrow} = (1 - \varepsilon_s)R_{lw}^{\downarrow} + \varepsilon_s \sigma T_g^4 \tag{9}$$

$$H_{sfc} = \rho c_p \left[\theta_g(\varepsilon_s) - \theta_a\right] r_h \tag{10}$$

where $\sigma = 5.67 \times 10^{-8}\ \mathrm{W \cdot m^{-2} \cdot K^{-4}}$, $T_g$ is the surface temperature, and $R_{lw}^{\uparrow}$ and $R_{lw}^{\downarrow}$ are upward longwave radiation and downward longwave radiation, respectively. The $c_p$ is the specific heat capacity of the air at constant pressure. According to the theory of heat transfer, the sensible heat flux (positive upward) must be consistent with the difference between ground temperature and air temperature. Under neutral conditions, if the emissivity is overestimated or underestimated, there is a difference between the estimated sensible heat flux and the observed sensible heat flux. Therefore, the surface emissivity of the KLML station is determined by minimizing the root-mean square difference (RMS) between the sensed heat flux estimate and the observed value.

### 2.2.3. Bulk Transfer Coefficients for Momentum (C$_D$) and Heat (C$_H$)

The bulk transfer coefficients for momentum (C$_D$) and heat (C$_H$) characterize the dynamic and thermal effects of turbulence, respectively, and are physical quantities that measure the strength of turbulence. Near-surface fluxes and heat fluxes can generally be expressed as follows:

$$\tau = \rho u_*{}^2 = \rho c_d u^2 \tag{11}$$

$$H = \rho c_p \overline{W'T'} = \rho c_p c_h (\theta_s - \theta_a)u \tag{12}$$

where $\rho$ is air density (kg/m$^3$), $u$ is horizontal wind speed (m/s), $u_*$ is friction velocity (m/s), $W'$ and $T'$ are the pulsation values of vertical wind speed and air temperature, $c_p$ is the constant pressure specific heat capacity of air (J/K·kg), $H$ is the sensible heat flux (W/m$^2$), and $\theta_s$ and $\theta_a$, respectively, are air temperature and surface temperature (K). In this paper, the data of eddy-related systematic observation are calculated by the following two formulas for C$_D$ and C$_H$:

$$c_d = (u_*/u)^2 \tag{13}$$

$$c_h = \frac{H}{\rho c_p (\theta_s - \theta_a) u} \tag{14}$$

Among them, the air temperature and surface temperature can be calculated from conventional data and long-wave radiation data, respectively.

*2.3. Remote Sensing Retrieval Aerodynamic Roughness*

In this paper, the Massman model is used to invert the aerodynamic roughness of the surface in the Gurbantunggut Desert area. The calculation formula is as follows [19]:

$$\frac{u_*}{u} = C_1 - C_2 e^{-C_3 C_d LAI} \tag{15}$$

$$n = \frac{C_d LAI}{2(u_*/u)^2} \tag{16}$$

$$\frac{d}{z} = 1 - \frac{1}{2n(1 - e^{-2n})} \tag{17}$$

$$z_{om} = h\left(1 - \frac{d}{z}\right) e^{-ku/u_*} \times f_a + (a \ln(EVI) + b)h \times (1 - f_a) \tag{18}$$

where $C_1$, $C_2$, and $C_3$ are 0.32, 0.264, and 15.1, respectively, and $C_d = 0.2$ is the drag coefficient of vegetation. LAI is calculated from the NDVI of MODIS. $n$ is the extinction coefficient of the wind velocity profile in the canopy; $a = 0.104$, $b = 0.31$, and h is the vegetation height.

## 3. Results

*3.1. Surface Roughness*

3.1.1. Aerodynamic Roughness

Based on the measured data of the KLML station, the $\ln z_{om}$ dataset is generated. Due to the observation error in the input measured data or the meteorological conditions that do not meet similar theories, the calculated $\ln z_{om}$ data are not a single value, but correspond to a value every half hour. Therefore, the optimal value of the $z_{om}$ will correspond to the peak in the $\ln z_{om}$ frequency histogram, i.e., the most frequent $\ln z_{om}$ can be used to calculate the $z_{om}$ at which the kinetic roughness is the optimal value for that period. For example, Figure 1a shows the frequency distribution of the KLML station $\ln z_{om}$ with a box width of 0.2; $\ln z_{om} = -4.5$ has the maximum frequency in the smooth curve, so the optimal value of $z_{om}$ is $1.11 \times 10^{-2}$ m. The morphological distribution and spatial distribution density of the underlying rough element are the main factors affecting aerodynamic roughness [20], and the roughness of the underlying surface with vegetation will vary significantly because of the change of plant height and cover density of different vegetation types; that is, it is proportional to the height of the vegetation canopy.

As shown in Figure 2b, the roughness of the KLML station has increased rapidly since the short-lived vegetation began to grow in April; from March to June it was $4 \times 10^{-3}$ m, $1.3 \times 10^{-2}$ m, $1.6 \times 10^{-2}$ m, and $2 \times 10^{-2}$ m, and June was also the maximum roughness of the year.

The annual aerodynamic roughness of the KLML station was $1.1 \times 10^{-2}$ m, which was higher than that obtained by Liu et al. [17] in the hinterland of the Taklimakan Desert of $3.1 \times 10^{-3}$ m. He Qing et al. [21] calculated the average roughness of Xiaotang Station to be $6.05 \times 10^{-5}$ m, Chen et al. [22] calculated the aerodynamic roughness of Gansu Jinta Oasis Desert Station to be $2.8 \times 10^{-2}$ m, and Yang Xinhua et al. [23] calculated the roughness of Hade Station in the northern edge of the Taklimakan Desert to be $2.70 \times 10^{-5}$ m, which is lower than that of Zhao et al. [24]. The overall dynamic roughness of Shiquanhe Station on the Qinghai-Tibet Plateau is $5.8 \times 10^{-2}$ m. Wang et al. [12] calculated and compared

the dynamic roughness of different types of underlying surfaces in different regions of the Qinghai-Tibet Plateau and found that they were related to the vegetation, topography, soil state, and seasonal differences of the underlying surface.

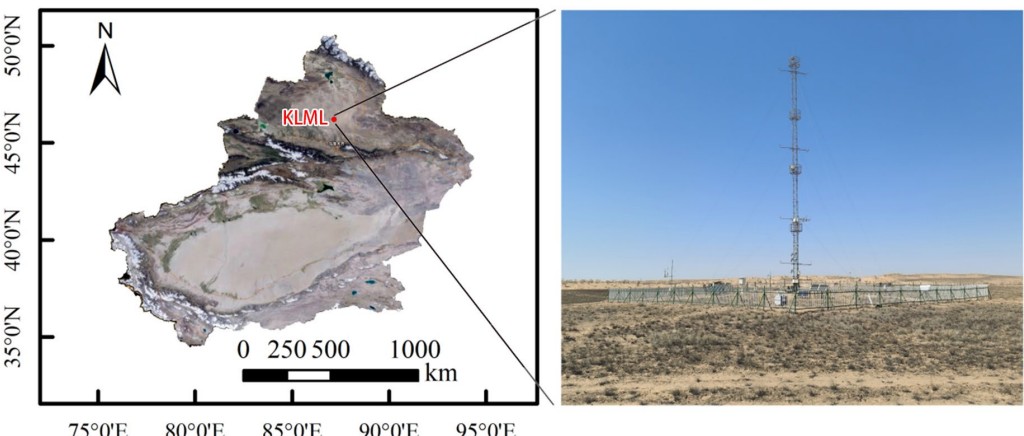

**Figure 1.** Map of KLML station at Gurbantunggut Desert.

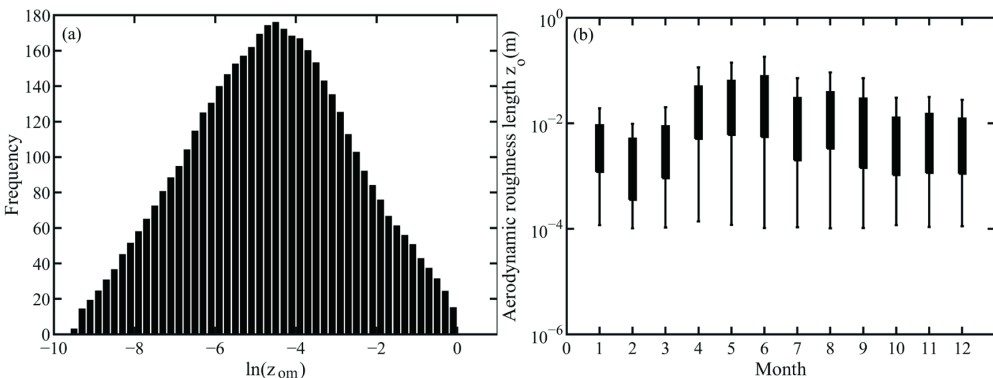

**Figure 2.** (**a**) $\ln z_{om}$ frequency distribution; (**b**) monthly variation in aerodynamic roughness.

Because the short-lived vegetation in June was withered, and the shrub growth increased the surface roughness, then the surface roughness reached the maximum of the year [25], which was much greater than other periods, and the difference could reach an order of magnitude.

For example, the difference between June and March is $1.6 \times 10^{-2}$ m, which indicates that the height and density of vegetation have an important influence on aerodynamic roughness and determine its magnitude. The Gurbantunggut Desert has a stable snow cover in winter, while the snow roughness is small, so the winter dynamic roughness is lowest throughout the year. In the process of vegetation growth to wilt from March to August, its change is affected by the flexibility of vegetation roughness elements; these elements are sensitive to wind speed. For all surfaces, the roughness under weak wind is larger, and the roughness under strong wind is small. As shown in Figure 3, the relationship between roughness and wind speed is that from April to September, as the wind speed increases from 1 m/s to 5 m/s, the roughness gradually decreases and tends to stabilize. Xia et al. [26] found through wind tunnel experiments that dynamic roughness is inversely proportional to wind speed regardless of vegetation cover on the underlying surface. Lu et al. in a Xiaotangshan experiment found that the dynamic roughness is related to the source region because of the heterogeneity of the roughness element size and spatial distribution in the source zone of different winds [27].

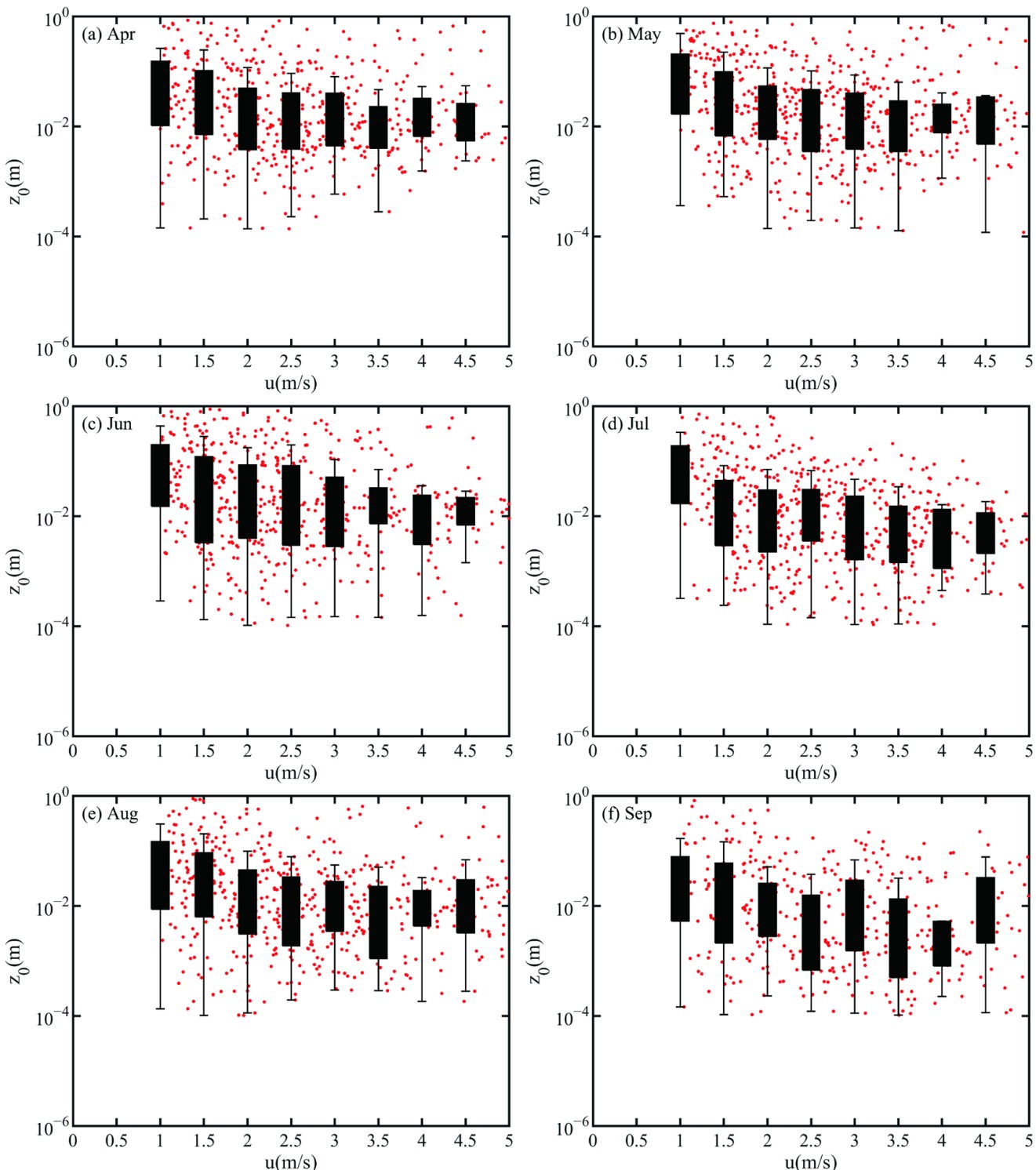

**Figure 3.** Relationship between wind speed and roughness at KLML Station in (**a**) April, (**b**) May, (**c**) June, (**d**) July, (**e**) August, and (**f**) September.

### 3.1.2. Thermal Roughness Length

In this paper, $kB^{-1}$ [$=\ln(z_{om}/z_{oh})$] is used to represent the thermal roughness change [14], and it can be seen in Figure 4 that the average daily change of $kB^{-1}$ January in all months has a significant daily variation, and the trend of change is similar to a parabola. $kB^{-1}$ gradually increases from about 7 a.m. to peak at noon, with peaks in April, May, and June occurring at 1 p.m., while peaks in July and August are at 3 p.m. In general, rough

surfaces have a higher $kB^{-1}$ than smooth surfaces because surface roughness affects kinetic roughness more than thermodynamic roughness. The trend of $kB^{-1}$ is similar to kinetic roughness, and the vegetation growth at the site in May and June is coarser than that in July and August, when the vegetation withers, so $kB^{-1}$ is higher than that in July and August.

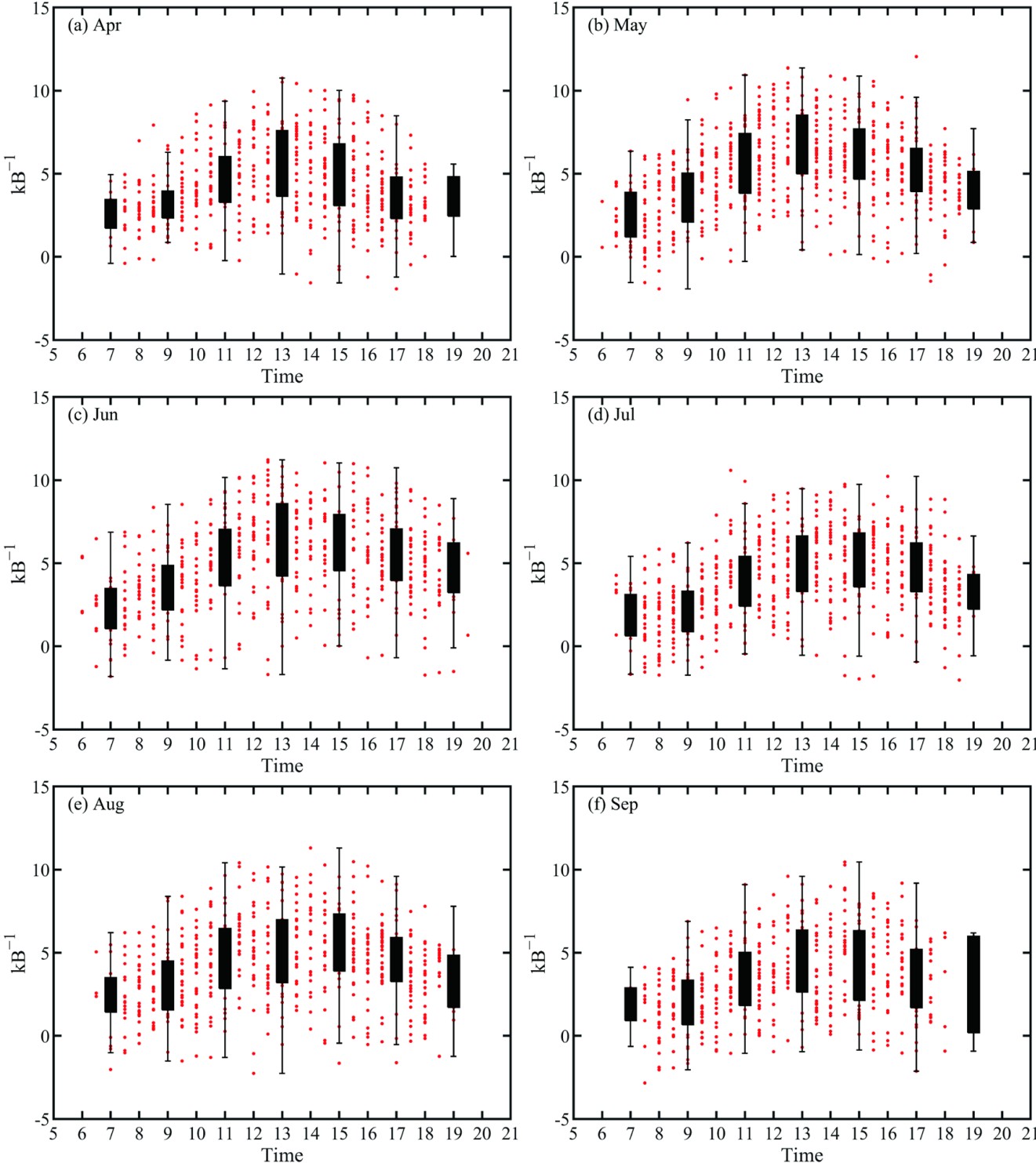

**Figure 4.** $kB^{-1}$ day change at KLML Station: (**a**) April; (**b**) May; (**c**) June; (**d**) July; (**e**) August; (**f**) September.

### 3.2. Surface Emissivity

By calculating the RMS difference between the directly observed sensible heat flux and the parameterized sensible heat flux at the KLML station, the given ratio of net radiation is determined at the time with the minimum RMS difference. As shown in Figure 5a, the ratio of net radiation is 0.93 when the RMS is the smallest. Figure 5b shows the monthly variation of the ratio of net radiation, with the highest value of 0.95–0.96 occurring during the winter when the surface is covered with snow and increasing with vegetation growth from April to May before decreasing with the withering of the vegetation. The ratio decreases to 0.91 in March–April and increases to 0.92 in June because of the strong growth of shrubs. After July, the ratio decreases again with the withering of vegetation. The ratio of net radiation increases with the increase in snow in the winter. Wang et al. [11] used NASA satellite data to analyze the spatial and temporal distribution characteristics of China's surface emissivity, and compared with the whole country, the Xinjiang desert area belongs to the region with the smallest specific emissivity (0.6163~0.9638); the results obtained in this article confirm this conclusion.

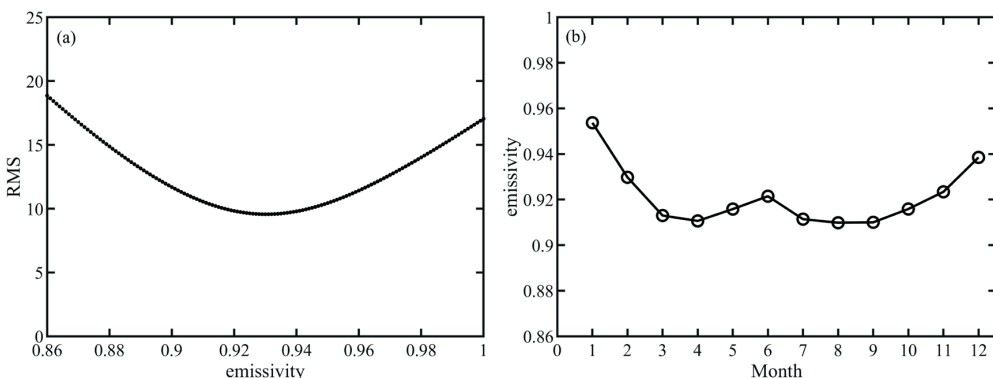

**Figure 5.** (**a**) RMS relationship between surface emissivity and sensible heat flux observations–parameterized sensible heat flux; (**b**) monthly change in surface emissivity.

### 3.3. Bulk Transfer Coefficients for Momentum ($C_D$) and Heat ($C_H$)

For unstable atmospheric conditions, when wind speeds are low, the main atmospheric turbulence is thermal turbulence, causing the upper part of the near-surface layer to be in a locally free convective state. This situation usually results in higher bulk transfer coefficients for momentum ($C_D$) and heat ($C_H$), as shown in Figure 6a,b. The values for both $C_D$ and $C_H$ are high in low wind speeds. As wind speed increases, turbulence shear is generated, and $C_D$ rapidly decreases with wind speed. When wind speed exceeds 6 m/s, the variation of $C_D$ decreases with wind speed. Figure 6c,d show that the $C_H$ also decreases with increasing wind speed, but the decrease is less than that for $C_D$. In this study, when z/L is greater than 0, the $C_D$ is greater than the $C_H$, with most of the $C_D$ ranging from $1 \times 10^{-1}$ to $1 \times 10^{-3}$, while the $C_H$ is less than $1 \times 10^{-3}$ when wind speed is greater than 5 m/s.

For stable atmospheric conditions, both $C_D$ and $C_H$ decrease with increasing wind speed at low wind speeds. As wind speed increases, the near-surface layer becomes closer to neutral conditions, and the increase in wind shear enhances mechanical turbulence, weakening the temperature inversion. The $C_D$ and $C_H$ remain relatively stable. Figure 7e–h show the relationship between bulk transfer coefficients and stability. It can be clearly seen that when stability is less than 0, the values of both $C_D$ and $C_H$ are significantly higher than when stability is greater than 0. $C_D$ are all above $3 \times 10^{-3}$, and $C_H$ are above $1 \times 10^{-3}$. While the $C_D$ decreases with unstable conditions changing from unstable to stable, it does not significantly change with the increase in unstable conditions from weakly unstable to strongly unstable. The relationship between $C_H$ and stability is similar, but the variation range is larger. When stability is less than 0, its value is mostly above $1 \times 10^{-3}$, but when stability is greater than 0, the $C_H$ is concentrated between $1 \times 10^{-4}$ and $1.6 \times 10^{-3}$. Furthermore, the $C_H$ increases

with increasing instability, but under stable conditions, the change in $C_H$ with stability is not significant. Yue [28] concluded that the momentum transport coefficient is greater than the heat transport coefficient under all stable conditions, which may be due to the different underlying vegetation conditions during the observation data period.

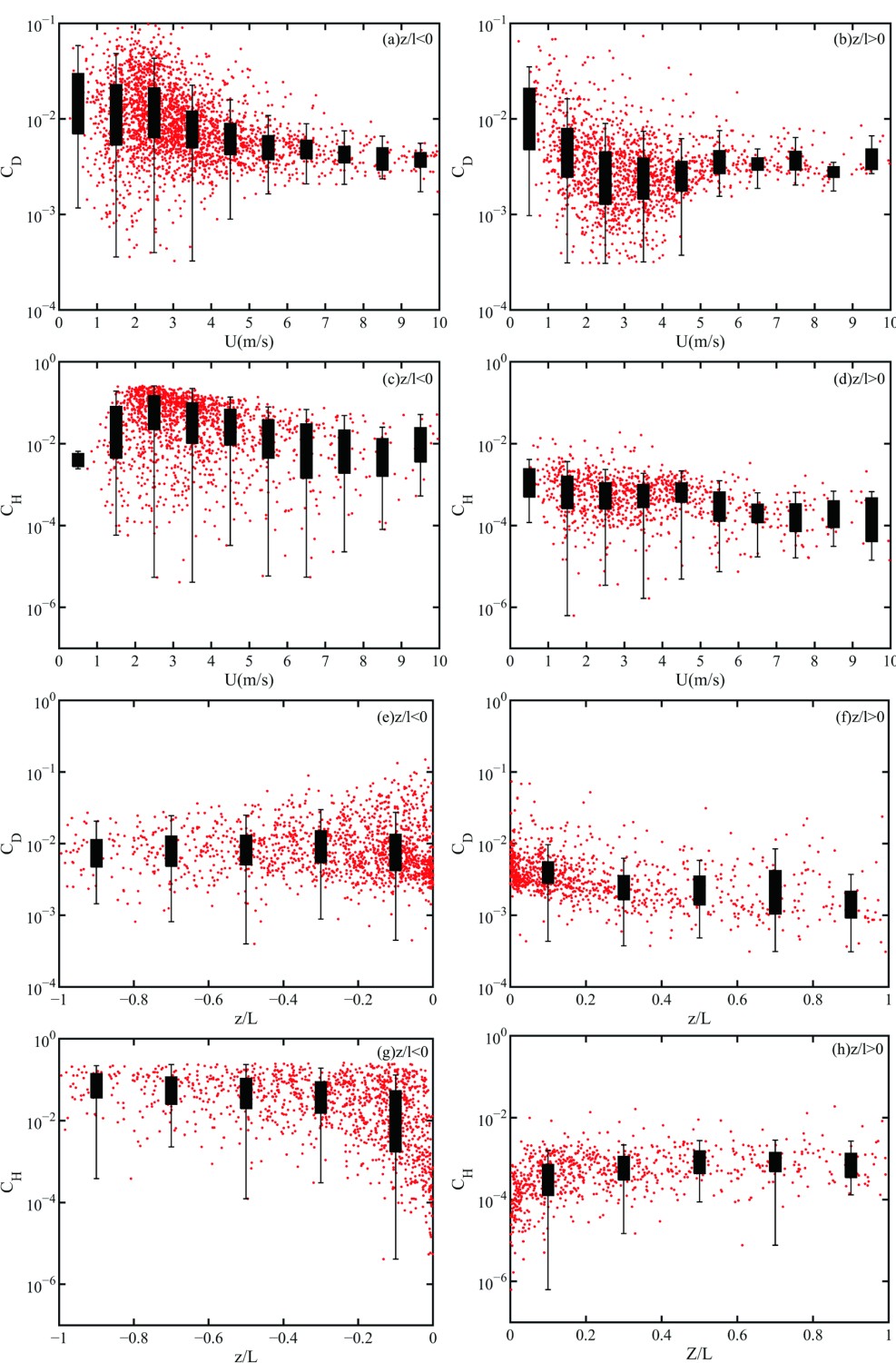

**Figure 6.** Relationship between bulk transfer coefficients for momentum ($C_D$) and wind speed: (**a**) z/l < 0, (**b**) z/l > 0; relationship between bulk transfer coefficients for heat ($C_H$) and wind speed: (**c**) z/l < 0, (**d**) z/l > 0; relationship between $C_D$ and z/l: (**e**) z/l < 0, (**f**) z/l > 0; relationship between $C_H$ and z/l: (**g**) z/l < 0, (**h**) z/l > 0.

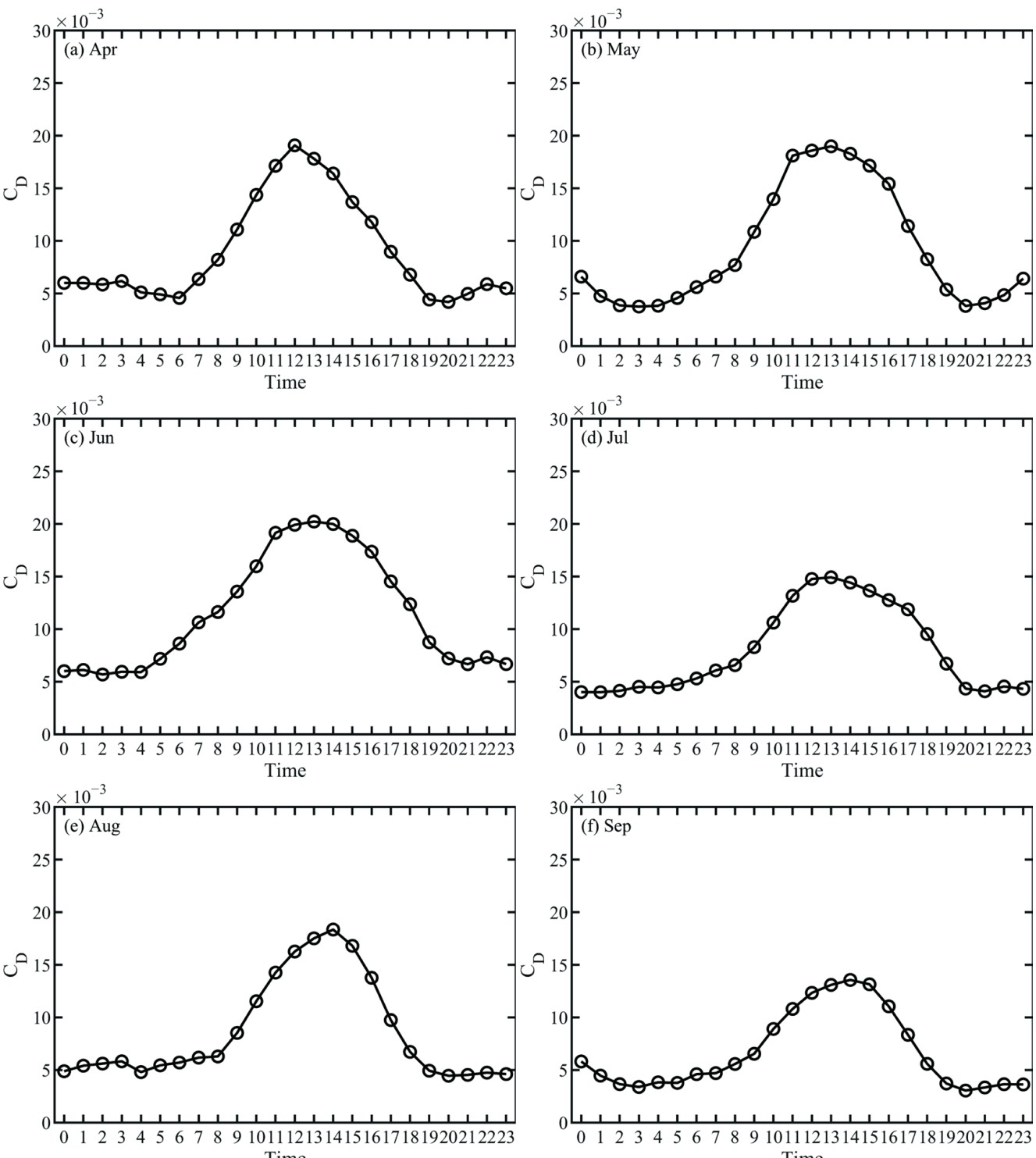

**Figure 7.** Monthly average daily variation of $C_D$: (**a**) April; (**b**) May; (**c**) June; (**d**) July; (**e**) August; (**f**) September.

Figures 7 and 8 give the monthly average daily change in $C_D$ and $C_H$ of KLML station from April to September. Whether it is $C_D$ or $C_H$, the daily average daily difference of monthly change is relatively large. The monthly average daily change trend of $C_D$ is high at noon, low in the morning and evening, starting to increase at about 6 a.m., reaching a maximum at 3 p.m., and dropping to a lower value at 8 p.m. The difference is that the $C_D$ will increase slightly after 8 p.m. in April and May. June–September is in a stable state

after 8 p.m. Overall, the June momentum transmission coefficient is the highest, when the peak is $2.02 \times 10^{-2}$. The September $C_D$ is the lowest, when the peak is $1.35 \times 10^{-2}$. The April–June value is higher than July–September.

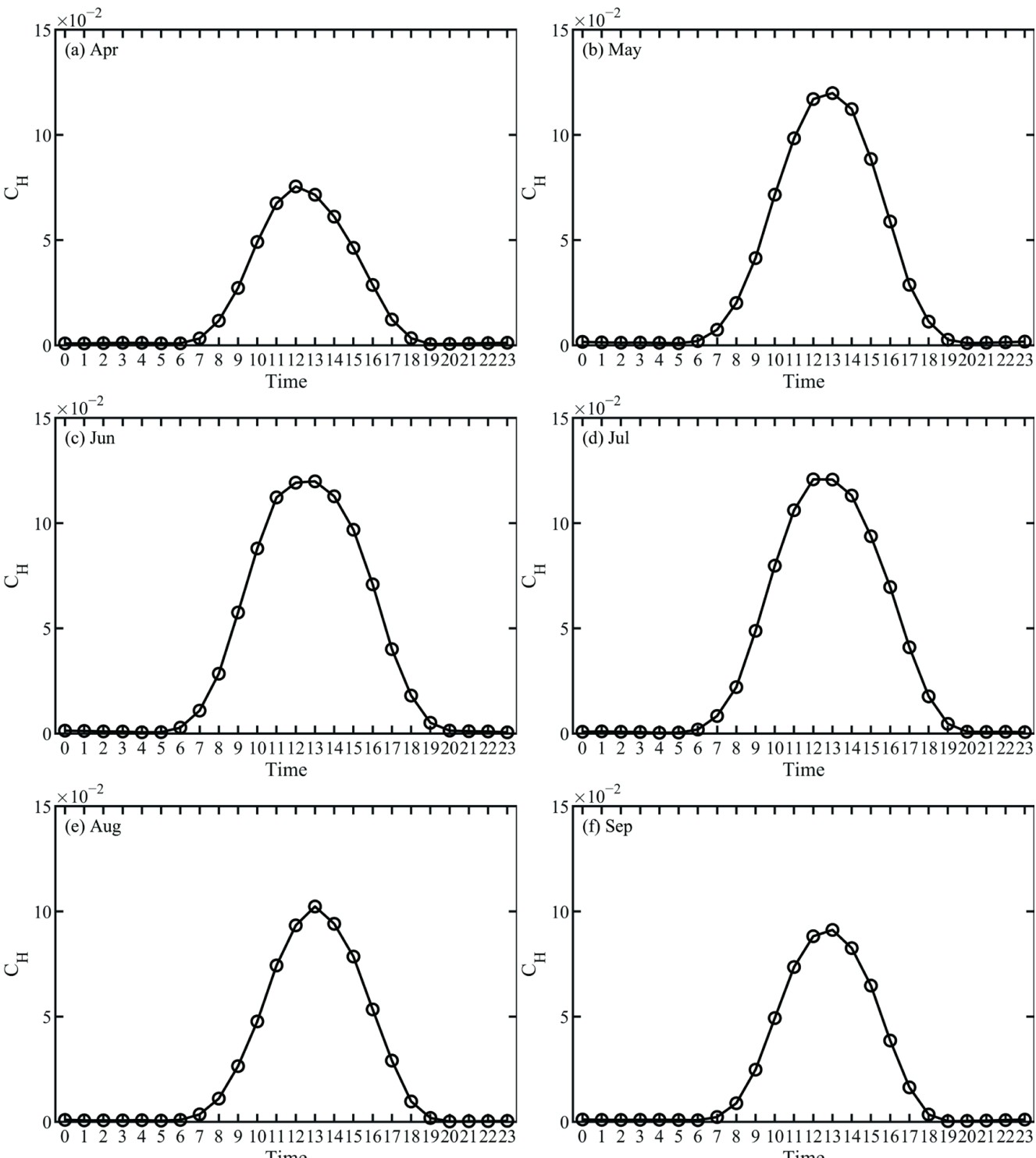

**Figure 8.** Monthly average daily variation of $C_H$: (**a**) April; (**b**) May; (**c**) June; (**d**) July; (**e**) August; (**f**) September.

The monthly average daily variation of the $C_H$ is very different from the monthly average daily change of the $C_D$; first of all, the change amplitude of the $C_H$ is much higher than that of the $C_D$. Second, the seasonal variation of $C_H$ is also obvious, which is obviously high from May to July, followed by August to September, and lowest in April, and the time difference between seasons caused by the annual and seasonal changes of solar radiation is also more obvious. The $C_H$ changes very drastically before and after sunrise, when the atmosphere transitions from stable to unstable (7 a.m. to 11 a.m.), which is in sharp contrast to the gentle change of the $C_D$ during this period, and its $C_H$ reaches its maximum in April and May early, and August and September at the latest. After reaching the maximum value, the $C_H$ decreases rapidly; around 9 p.m., all months drop to the lowest. When the atmosphere stabilizes the stratum from 0:00 to 7:00 a.m., the difference in $C_H$ from April to September is not large, and it begins to increase at 6 a.m. Wang et al. [12] and Yang et al. [29] calculated the overall transport coefficient of multiple stations on the Qinghai-Tibet Plateau and found that the overall transport coefficient was greatly related to the observation period and the type of underlying surface. The momentum transport coefficient is high in summer and low in winter, and the average daily variation of each month is low at night and high during the day, and Zheng et al. [30] also reached the same conclusion in the Badain Jaran Desert study.

### 3.4. Remote Sensing Retrieval Aerodynamic Roughness

Figure 9 shows the aerodynamic roughness of remote sensing retrieval in the Gurbantunggut Desert area, which was $2.37 \times 10^{-2}$–$2.46 \times 10^{-2}$ m in the central desert from April to November, $1.41 \times 10^{-2}$–$2.04 \times 10^{-2}$ m in the northwest, and $1.53 \times 10^{-2}$–$2.39 \times 10^{-2}$ m in the east. The aerodynamic roughness of the Gurbantunggut Desert is higher in the middle than in the northwest and east because the northwest and east are mostly semi-fixed sand and semi-fixed flat sand, and vegetation is very rare. In March, the temperature of snow melt warms up, and in April–May, moisture is more abundant, short-lived vegetation growth accelerates, and roughness increases. May is the most vigorous period of short-lived vegetation, and roughness reaches the maximum of the year. Although short-lived plants withered in June, shrubs grew vigorously and roughness was still at a high value, and then the lack of water in July and November was no longer suitable for vegetation growth compared with April–May. The contribution of vegetation to roughness weakened, the height and coverage of surface vegetation gradually decreased, and then, the snow cover increased in winter, the roughness decreased, and the range of large value areas also decreased. Xing et al. [31] compared the relationship between different vegetation indices and aerodynamic roughness and found that NDVI is a sensitive indicator of grassland aerodynamic roughness. Abbas et al. [32] obtained the best final mathematical model describing the relationship between NDVI and aerodynamic roughness for plotting the length of roughness throughout Iraq. Cho et al. [33] used the ratio of LAI to canopy height to characterize the complexity of canopy structure to examine the relationship between aerodynamic roughness and albedo. Sun et al. [34] used the empirical model and LAI data developed by predecessors to study the spatial distribution of aerodynamic roughness at small spatial scales in the northern Qinghai-Tibet Plateau. Liu [35] et al. used turbulent transport models to identify canopy morphological characteristics related to LAI and plant functional types. LAI quantifies the leafy area of terrestrial vegetation, which is a fundamental attribute of vegetation canopy structure and function [36]. Surface dynamic roughness not only changes with time scale but also with spatial scale, and the central and eastern deserts were at high values of $2.48 \times 10^{-2}$ m and $2.42 \times 10^{-2}$ m, respectively. The roughness value in the northwest is low, $2.04 \times 10^{-2}$ m, and the remaining months show a trend of high in the middle and low in the east.

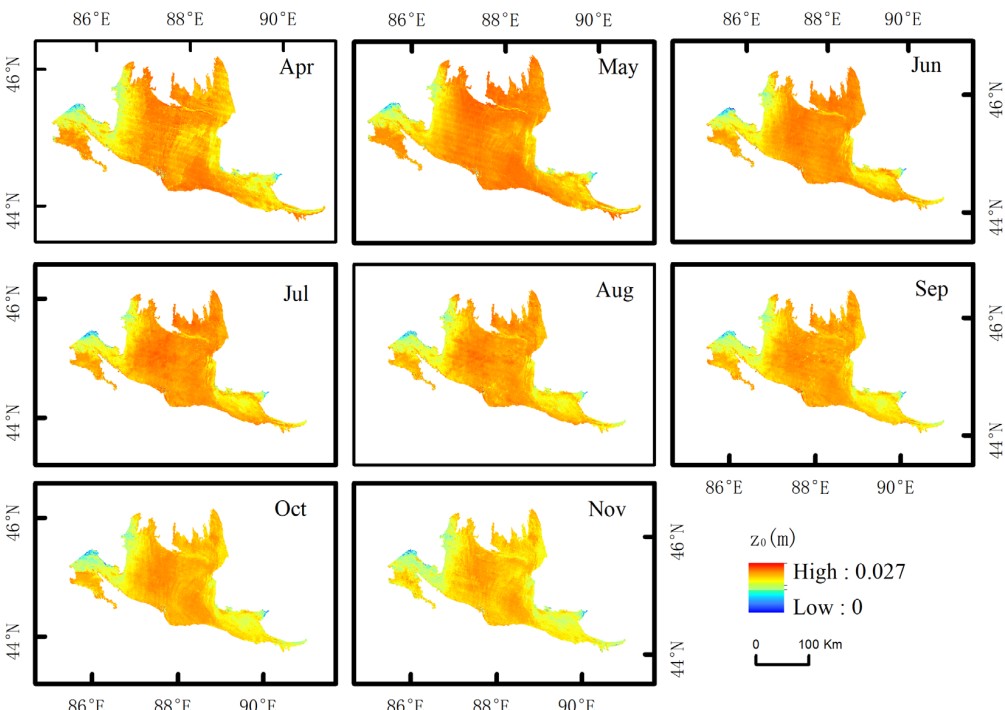

**Figure 9.** Remote-sensing retrieved aerodynamic roughness.

## 4. Discussion

In this paper, the key parameters of energy exchange (surface roughness, specific emissivity, and overall transport coefficient) in the land–air interaction are calculated by taking the hinterland of the Gurbantunggut Desert as the research object and comparing it with the research results of many scholars at home and abroad. The changes of surface parameters and their influencing factors in the underlying desert with short-term vegetation cover are discussed, which is helpful to deeply understand and recognize the land–air interaction law of the underlying surface of this type and the design of the atmospheric model parameterization scheme. The results of the study mainly reveal the following.

Due to the large roughness of the surface during the period of short-lived vegetation, the kinetic roughness of the KLML station from April to June was the highest throughout the year, and its change trend was related to the growth trend of vegetation; the vegetation tended to be vigorous and the kinetic roughness increased. Although the vegetation withered in June, its withered body was still left on the surface, and the surface roughness increased instead of decreased. The overall decrease in aerodynamic roughness decreases with the increase in wind speed and eventually stabilizes, which is the same as the conclusion obtained by Yu Mingzhao et al. [37] in the Heihe and Haihe River basins. Zhang et al. [38] also obtained the same result by comparing the changes of roughness of forests, farmland, and grasslands. The remote sensing retrieval results are different from the station calculation results, and the annual peak of the entire desert area macroscopically occurs in May, and the change trend is the same as the station results, which changes with the melting of snow, the amount of moisture, and the growth and wilting of vegetation, which is similar to the results obtained by Liu et al. in the Nagqu area of the northern Tibetan plateau. A large number of studies have shown that aerodynamic roughness is related to vegetation canopy structural parameters, so LAI has a stronger correlation than NDVI to some extent when inverting aerodynamic roughness, and zero-plane displacement height and aerodynamic roughness are strong functions of LAI [33,34,39]. In terms of spatial distribution, the aerodynamic roughness of the Gurbantunggut Desert was generally high in the middle and low in the east and west, and the change trend in time was the same as that calculated by the station, which increased from April to June and gradually decreased after July. The change trend of

$kB^{-1}$ is similar to kinetic roughness and has a clear diurnal trend, high at noon, low in the morning and evening, which is consistent with the conclusion of Wang et al. [12]. The maximum value of $kB^{-1}$ occurs at 1 p.m. in April–June, and at 3 p.m. in July–August.

The surface reflectivity changes with the change of land cover type and is closely related to the ratio of underlying vegetation to soil [40–42]. Ice and snow surfaces generally have a high specific emissivity. When the KLML station is covered with snow in winter, the specific emissivity is 0.95; the snow and ice begin to melt in spring, and the vegetation begins to grow, but the surface exposure ratio is still high, resulting in a decrease in the specific emissivity in March and April. By June, as the vegetation grows to its most vigorous, the specific emissivity increases, and after June, because of a lack of water, the vegetation gradually withers, and the specific emissivity decreases. Liu et al. [43] found that the surface reflectivity of the Taklimakan Desert was most affected by soil moisture, and the reflectivity was as high as 0.93 in the poplar forest area near the oasis, and the reflectivity in the arid desert center area was 0.90~0.91. Zhai et al. [44] found by analyzing the spatiotemporal situation of the specific emissivity of the land surface in China from 2000 to 2011 that the specific emissivity of bare soil is often lower than that of green vegetation, which is the reason why the specific emissivity of the KLML station from June to August is lower than that of May.

The $C_D$ and $C_H$ are inversely proportional to the wind speed, and the overall momentum transport coefficient is mainly concentrated in low wind speed because low wind speed is conducive to the development of unstable convection. When the wind speed is low, thermal turbulence is the main atmospheric turbulence movement, so that the upper part of the near formation is in a local free convection state, resulting in a large momentum and $C_H$ and turbulent shear. With the increase in wind speed, the transmission coefficient decreases rapidly, and gradually tends to neutral, which is the same as the research conclusion determined by Sun Jun et al. [25] in the Heihe River Basin. When the wind speed exceeds 6 m/s, the trend of momentum transmission coefficient decreases with the increase in wind speed and slows down and tends to be stable. When the stability is less than 0, the $C_D$ and $C_H$ are higher than the values when the stability is greater than 0, and the $C_D$ decreases with the increase in stability but does not change significantly with the increase in instability from weak instability to strong instability, which is consistent with the results of previous studies [12,17,25].

Compared with the existing research results, this paper uses the measured data of the vortex correlation system of the observation station, which is more accurate than the remote sensing data products, and the data interval is half an hour, which is more conducive to analyzing the daily changes of various parameters. However, the measured data will have poor quality or even lack some data because of unexpected situations such as power failures, and this kind of data can be excluded after screening, resulting in incomplete observation time series, which affects the calculation results to a certain extent. The advantage of remote sensing data inversion results is that it can obtain data quickly over a wide range, which is suitable for large-scale and long-term series research, and is more compatible with various pattern grids.

## 5. Conclusions

This article calculates key parameters in land–atmosphere interaction based on data from the KLML station and satellite data and analyzes the results. The following conclusions are drawn.

The aerodynamic roughness of the KLML station is $1.1 \times 10^{-2}$ m, which is affected by the change of the underlying surface, which increases from April to June, and then gradually decreases, changes with the growth trend of vegetation, decreases overall with the increase in wind speed, and tends to be stable. The annual trend of $kB^{-1}$ is similar to the kinetic roughness, with daily changes of midday high, morning and evening low, and the maximum value from 13:00 to 15:00. The aerodynamic roughness of the Gurbantunggut Desert was obtained by retrieval of MODIS data, and its temporal variation

trend was the same as that of the station results, which increased from April to June and gradually decreased after July because of the influence of snow melting and moisture and vegetation growth and wilt. The spatial distribution was positively correlated with vegetation cover, high in the central part and low in the eastern and western parts, and the aerodynamic roughness in the central part was $2.37 \times 10^{-2}$–$2.46 \times 10^{-2}$ m from April to November, and $1.53 \times 10^{-2}$–$2.39 \times 10^{-2}$ m and $1.41 \times 10^{-2}$–$2.04 \times 10^{-2}$ m in the east–west part, respectively.

The annual specific emissivity is 0.93; the specific emissivity changes with the change of snow and vegetation in the underlying area. In winter, because of the high specific emissivity of snow cover, the specific emissivity in January is 0.95. In the spring, ice and snow melt, the surface exposure ratio increases, the specific emissivity decreases, the shrub grows vigorously in June, the specific emissivity increases, and after June, because of a lack of water, vegetation gradually withers. The specific emissivity in July and August is lower than in June.

The $C_D$ and $C_H$ are inversely proportional to the wind speed. When the wind speed is lower than 6 m/s, the transmission coefficient decreases rapidly with the increase in wind speed, and when the wind speed exceeds 6 m/s, the trend of $C_D$ decreasing with the increase in wind speed slows down and tends to be stable. When the stability is less than 0, the $C_D$ and $C_H$ are higher than the values when the stability is greater than 0. The $C_D$ decreases with the increase in stability, high in summer and low in winter, and the average daily variation of each month is low at night and high during the day. The magnitude of the $C_H$ is higher than that of the $C_D$, and the change from 7 to 11 is very sharp, which is different from the gentle change of the $C_D$ during this period, followed by the $C_H$ in May–July being higher than that in August–September.

**Author Contributions:** Conceptualization, A.M. and W.L.; methodology, Y.L. and W.L.; software, W.L. and A.A.; validation, A.M., Y.L. and J.G.; formal analysis, W.L. and F.Y.; investigation, Y.W.; resources, W.W. and Z.C.; data curation, W.L. and W.H.; writing—original draft preparation, W.L.; writing—review and editing, W.L. and C.W.; visualization, W.L.; supervision, M.S.; project administration, C.Z.; funding acquisition, A.M. All authors have read and agreed to the published version of the manuscript.

**Funding:** This research was funded by the China Desert Meteorology Research Foundation (Sqj2021006), the Special Project for the Construction of Innovation Environment in the Autonomous Region (PT2203), the Scientific and Technological Innovation Team (Tianshan Innovation Team) project (Grant No. 2022TSYCTD0007), the S&T Development Fund of IDM (KJF202302), the National Natural Science Foundation of China (Grant No. 41875023), and the Special Funds for Basic Scientific Research Business Expenses of Central-level Public Welfare Scientific Research Institutes (IDM2021005, IDM2021001).

**Data Availability Statement:** The data used in this paper can be provided by A.M. (ail@idm.cn) upon request.

**Conflicts of Interest:** Author Wenbiao Wang and Zhengnan Cui were employed by the company Elion Resources Group Co. The remaining authors declare that the research was conducted in the absence of any commercial or financial relationships that could be construed as a potential conflict of interest.

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
