# Peer review of "Parameterization and Remote Sensing Retrieval of Land Surface Processes in the Gurbantunggut Desert, China"

_remotesensing, doi:10.3390/rs15102646_

Round 1

Reviewer 1 Report

human impact is significant. A unique natural object - Gurbantunggut Desert in northern Xinjiang, northwest China, is among such territories. The interest to this desert is also determined by the fact that it is equidistant and rather far from any sea. Therefore, the results obtained by the authors are urgent and of great interest to readers of the journal and specialists in the field of atmospheric physics.

The manuscript investigates the atmosphere-underlying surface interaction in interior areas of the Gurbantunggut Desert and compares the obtained results with the results of other investigators.  The key parameters of energy exchange, such as surface roughness, emissivity, bulk transfer coefficients, are examined. In their study, the authors use the data of both KLML station and remote sensing.  Thus, the study is fully consistent with the subject of the Remote Sensing journal.

I‘d recommend the authors to improve the presentation of their results, for example

1. Line 93.  The word "depth" is in a different font.

2. Lines 156, 172, and 255.  The words "surface," "bulk," and "surface" should be capitalized after the dot.

3. There is a lot of empty space on Page 6; it should be filled with text.

4. The figures design does not meet the style guide requirements.  The axes are labeled in small font, making it difficult to read.  Figure lettering (a), (b), etc. should be taken outside the figures. In Fig. 2a, Z should be replaced with z.  I’d recommend the authors to read their manuscript more carefully to avoid sloppiness when presenting the results. Remote Sensing is a highly rated journal and the presentation of results should be as close to ideal as possible.

5. In the manuscript, a lot of work is done to compare the results obtained with the results of other authors. The effect of anthropogenic factors on the environment of the Gurbantunggut Desert is also of great interest to a reader. It is known that a highway crossing the desert is currently under construction. Its impact on the desert environment is becoming increasingly significant. In addition to a current increase in the temperature observed in many regions of the world, a significant change in the wind speed also takes place.  According to Eq. (1), the surface roughness z0m depends on the wind profile.  It is very interesting whether a significant change in surface roughness z0m was observed after the construction of the highway in the Gurbantunggut Desert.  This information would be very useful to the readers.

I believe that this article should be published in the journal Remote Sensing.

Reviewer 2 Report

The manuscript is well worth to be published. Described below are my suggestions to improve the manuscript. Overall, the manuscript contained many long sentences. They should be separated into several shorter sentences. My specific comments are:

a) Line 38 - 'chiefly' can be changed to 'mainly'

b) line 38 - "Essential" can be changed to "The key"

c) line 56 - 'Guoyin' can be removed.

d) line 57 - 'SACOL' should be spelled out

e) line 60 - 'Xinsheng' can be deleted.

f) line 70-71 - 'Scholars ... and others' can be changed to 'Several researchers'

g) Lines 78-85: This is a very long sentence. It should be re-written and broken up into several shorter sentences.

h) line 91 - ', and is' can be changed to '. It is'

i) Subsection 2.2.2. surface emissivity - Have you ever compared your calculated surface emissivity with those from EOS ASTER surface emissivities or MODIS surface emissivities?

j) the Discussion section is too long. Most of the materials should be moved to the previous section.

Many sentences in the manuscript are too long. They should be separated into several sentences.
